# Why They Stayed and Why They Left—A Case Study from Ellicott City, MD after Flash Flooding

**DOI:** 10.3390/ijerph191710636

**Published:** 2022-08-26

**Authors:** Alisha Yee Chan, Kate Burrows, Michelle L. Bell

**Affiliations:** 1Chemical and Environmental Engineering, Yale University, New Haven, CT 06511, USA; 2Institute at Brown for Environment and Society, Brown University, Providence, RI 02903, USA; 3School of the Environment, Yale University, New Haven, CT 06511, USA

**Keywords:** flash flooding, phenomenological analysis, sense of place, relocation

## Abstract

Ellicott City, MD was devasted by flash flooding in 2016 and 2018. A lack of qualitative research has been conducted on topics related to sense of place and flash flooding, especially in the United States. In this study, we reveal reasons why some who experienced flash flooding continued to stay the flood zone and why some leave. We utilized a phenomenological approach to answer these research questions. Data were generated through in-depth interviews with 19 participants from the Historic District and adjacent neighborhoods in Ellicott City. The most common reasons participants stayed were: (1) Community Impact, (2) Historical Land, and (3) Financial Burden. The most common reasons participants left were: (1) Emotional Exhaustion and Frustration, (2) Fear/Anxiety, and (3) Financial Burden. The results of our study indicate that reasons individuals who experience flash flooding stay, or leave may include community/historical, environmental, emotional, and economic factors. This reveals the complexity of relocation and sense of place after natural/environmental disasters and supports previous literature that suggests tailored response efforts based on these unique set of burdens. This paper aims to identify burdens and understand flood victims’ decisions to help policy makers improve flood response efforts.

## 1. Introduction

Ellicott City, MD was devasted by two rare large flash floods, one in 2016 and another in 2018 [1,2]. Homes, small local businesses, vehicles, possessions, and lives were lost. The Historic District, also known as Main Street or Old Ellicott City, and the nearby neighborhoods suffered the most damage from these flash floods [1]. Approximately 71 people were displaced after the first flash flood in 2016 and over 60 were displaced after the second in 2018 [3,4].

Flash floods in urban areas can occur with only minutes to a few hours of heavy rainfall, potentially devasting lives and infrastructure along their path [5]. Flash flooding is expected to have different health, economic, and destruction impacts than overbank flooding because of its sudden and unexpected onset [5].

According to previously published literature, individuals who unwillingly relocated after flooding reported not only losing their homes, but also their sense of place and self-identity [6,7]. Previous literature on sense of place has described it as a person’s or group’s meaningful relationship with their environment [8,9,10,11,12]. Sense of place is largely impacted by an individual’s unique history, values, emotions, beliefs, memories, and experiences [13]. Existing qualitative literature on the lived experiences of individuals who experienced flooding have revealed themes such as anxiety during rainfall, changes in interpersonal relationships, loss of place of employment, mental health issues, and community altruism [14,15,16,17,18,19,20]. As we discuss throughout this paper, these factors often influenced whether individuals stayed or relocated after flash flooding.

In the United States, 99% of counties were impacted by at least one flooding event between 1995–2019 [21]. Further, flash flooding is one of the leading causes of death from natural/environmental disasters in the United States [22]. Despite this, the number of studies that have qualitatively reported the experiences of individuals impacted by flash flooding through phenomenological studies is limited, especially within the United States [14,15,17,18,20,23]. Research on flash flooding is of the upmost importance given the increase in severity of storm events from climate change and the rising installation of impervious surfaces from urbanization. Given these concurrent trends of climate change and urbanization, flash floods may increase in strength and frequency and devastate more lives in the future [24].

Given the growing importance of flash flood research, we investigated the experiences of flash flooding in Ellicott City, MD. In particular, we examined why some residents and employees continued to stay in Ellicott City’s flooded areas while others relocated.

## 2. Methods

We applied phenomenological research methods with detailed in-depth interviews of persons who experienced the flash floods in Ellicott City.

### 2.1. Study Site

Ellicott City (Figure 1) is located in Howard County, MD and as of 2020, has a growing population of over 75,000 people [25]. Ellicott City has a high median household income of $131,534 [25]. However, further analysis at the Census block group level shows that the Census block group that contains our study site, the Historic District of Ellicott City, has a median household income of $62,727, less than half of that of the entire Ellicott City (based on American Community Survey’s 2015–2019 5-year estimates). Ellicott City is located approximately 13 miles west of Baltimore, MD and 37 miles northeast of Washington DC. The Historic District resides within the Patapsco Valley in the Tiber Branch watershed and has experienced documented overbank flooding dating back to at least 1768. However, in 2016 and 2018, multiple tributaries that drained towards Ellicott City overflowed and resulted in major flash floods [1]. As of 2019, 65.2% of the Tiber Branch watershed was developed land, and 30.9% of that land consisted of impervious surfaces [1]. The increase in the percentage of developed land and impervious surfaces within the Tiber Branch watershed as well as the existing steep stream slopes largely contributed to the sudden flash flood events that occurred in 2016 and 2018 [1]. The Historic District of Ellicott City is a unique study site because it contains National Historic Landmarks such as the oldest surviving train station in the United States and buildings that have stood for hundreds of years. Ellicott City’s Safe and Sound plan in 2019 aimed to support local businesses, residents, and prepare for climate change events including flash flooding [26].

### 2.2. Participants and Interviewing Strategy

Participants were recruited through a snowball sampling approach between April 2020 to August 2020. The advertisement for study participants was posted on an Ellicott City Facebook group and shared or reposted by the members of the group. At the end of each interview, the interviewer asked the participant to share the flyer with others who experienced the Ellicott City floods. The primary inclusion criteria were respondents that either lived or worked in Ellicott City during one or both flash floods. All participants were required to be 18 years of age or older at the time of the interview. We conducted 19 in-depth interviews (each lasting about 1 h), which falls within the recommended sample size for phenomenological research ranging from 1–25 participants [28,29,30]. Two couples were individually interviewed during this study (4 participants). Partners were deemed acceptable separate interview participants if they were not in the same location during the flood(s). For example: one participant was at their business on Main Street during the 2016 flood but was not present during the 2018 flood while their spouse was present during the 2018, but not the 2016 flood. These individuals were both interviewed. A USD 25 Amazon gift card incentive was provided to the participants after the interview in compensation for their time. Respondents provided informed verbal consent before being interviewed. Ethical approval for this study was granted by the Yale Institutional Review Board. All identifying information was removed and each interview was assigned a randomly auto-generated four-digit number. This anonymous four-digit identifier is shown in parentheses at the end of participant quotes throughout the results section. Interviews were transcribed using a paid external service, VerbalInk.

Due to the COVID-19 pandemic, the study was advertised only through social media platforms and interviews were conducted by the first author through video conferencing. We used a semi-structured interview guide to conduct interviews and initiate discussion about the experiences of the participants regarding the floods. The interview guide was used to frame conversation, but we did not adhere to it rigidly during the interview, as is appropriate for a phenomenological research [31]. Though these interviews were conducted two years after the second flood, participants were able to recall events that occurred years ago. Previous research suggests that victims of a disaster have the ability to describe detailed and accurate accounts of events even six years after the disaster [32].

### 2.3. Phenomenological Analysis

Analysis was performed using phenomenological methods [33] to answer the research questions: why did participants stay in Ellicott City’s flood zones and why did participants leave? De-identified transcriptions were first thoroughly read and significant phrases were extracted, during which the researchers identified reoccurring themes related to the research question based on the extracted phrases. Themes were then organized into a clearly understood narrative. The researchers then returned to the original transcripts to extract quotes/phrases that provided evidence to support or contradict the themes, during which time the researchers identified additional reoccurring themes and the data analysis process was repeated and iterated until themes and supporting quotes were identified. The objective of this method of analysis was to allow themes to be formulated based on the evidence or quotes from the interviews and ensure that these themes accurately represented the findings from the interviews and were not retrofitted to fit a pre-structured narrative. Previous studies have used this approach to investigate community responses and perspectives in relation to environmental disasters [28].

## 3. Results

Table 1 shows that 14 out of 16 of the participants who lived in Ellicott City’s flood zone during the flash floods still live in the flood zone in Ellicott City at the time of the interview while the remainder relocated. One participant spoke extensively about their plans to relocate soon after the time of the interview. Though they were not counted as “having relocated” in Table 1, their statements about relocation were included and in the results of this study. In turn, only 4 out of 8 of the participants who worked in or owned a small local business in Ellicott City’s flood zone during the flash floods still did so at the time of the interview.

### 3.1. Why Do They Stay?

Three essential themes were identified relating to the research question-why do the individuals who experienced the flood(s) of Ellicott City stay after experiencing flash flooding? The identified themes were: (1) Community Impact, (2) Historical Land, and (3) Financial Burdens. Most participants described the Historic District as “a great community…[with] historic charm. The people are really friendly. It’s a great place to live” (9501). However, as we will see in the financial burdens section, not all participants had stayed willingly.

#### 3.1.1. Theme 1: Community Impact

Almost all participants who stayed in Ellicott City’s flood zone reported that they did so because of the “tightknit community” (7395) and the strong bonds of community cohesion that were “strengthened through the floods” (4282). For example, a participant described building strong friendships with fellow residents who experienced flash flooding.

“When you go through something traumatic…you make friends out of distress and those friendships are deeper and more meaningful…you know, I’m helping people throw out their only copies of their children’s baby photos…I think it’s a level of intimacy that you usually only have with close friends.”(8503)

Many participants reported seeking solace in their fellow community members. Among the participants that had stayed, most had reported that it was because of their supportive and empathetic community that they know understands what they’ve been through.

“The biggest, the reason why we stay is that…there’s something to be said about people who have been through the same experience that you have. We all have PTSD here—we do. We know it, and folks who know what it is, who have been through it as well, they understand… And it’s a support network, and emotionally, for me, it was important to stay.”(7489)

Participants also described feelings of reassurance of future flood safety through trusted community members and neighbors. One participant said, “I know if something happens, I’m not going to be alone…I’m going to help [my community] and they’re going to help me” (4829). Others were grateful for the support from their local church, and trusted that that support would be there for any future flood-related events:

“Having the church there available has been fantastic. Knowing that it’s there next time is I think a real relief for a lot of people. It’ll at least give us a place to go, you know?” (4455).

#### 3.1.2. Theme 2: Historical Land

Throughout our interviews with participants, it became apparent that the historical attributes of our study site are important characteristics for many who live(d)/work(ed) in the Historic District. A few participants had credited the reason for their unwillingness to relocate to the town’s unique “historical center that got national recognition” (4720). A participant had expressed that this historical treasure would be “hard to let go [of] and not try to build it back up” (4720). One participant proudly described their house as being a hundred years old and the foundation as being 200 years old (8392). Another participant reported that their great, great grandparents got married in the old church in Ellicott City, revealing that the Historic District is not only contains national history, but also contains “family history” (1247) for many.

#### 3.1.3. Theme 3: Financial Burdens

However, not all participants that stayed reported doing so willingly. Two participants had expressed their desire to relocate but could not because “FEMA will give [us] money to rebuild exactly the way [we] were, but maybe FEMA should give [us] money to move…the incentive is to stay put, and then get reflooded and reflooded and reflooded” (4455).

### 3.2. Why Do They Leave?

Three essential themes were identified relating to the second research question-why did individuals of Ellicott City who experienced flash flooding leave? The reoccurring themes were (1) Emotional Exhaustion and Frustration, (2) Fear/Anxiety, and (3) Financial Burden.

#### 3.2.1. Theme 1: Emotional Exhaustion and Frustration

Multiple participants had distanced themselves from the community due to emotional exhaustion after experiencing multiple flash floods. One participant, who still lived in the Historic District at the time of interview but had definitive plans to relocate, said:

“I don’t talk to a lot of the homeowners, the renters, the business owners anymore. I used to be very close with them, so in 2016, it was very devastating because I knew all these people. I visited their houses, and I hung out with them…so the 2016 flood was very devasting just being in this community, seeing how it just broke everyone…I have a sense of detachment I think from the town now because I don’t want to be fully invested anymore because it’s exhausting going through two [floods]…I don’t want to be emotionally invested in this town anymore, or at least Old Ellicott City anymore, because I don’t want to have to feel that again.”(7395)

Many participants also expressed worry about the anticipated effects of climate change and frustration with the government for continuing to allow further development. Further, some felt that government stormwater mitigation initiatives were inadequate and too slow-paced. When we asked a participant why they did not reopen their business, they responded that they fear that another flash flood will occur because “of what’s going on around the world with global warming…and [impervious] development” (8303). The participant angrily stated that, “all [the government has] done is talk, talk, talk since 2016…to this day, they haven’t really done anything” (8303).

#### 3.2.2. Theme 2: Fear/Anxiety

The participants who still live or work in the Historic District all explicitly described feelings of fear and anxiety during rainfall. Almost all participants reported that they believe there will be future flash flooding in Ellicott City. Some participants seemed very certain that another would occur, using phrases such as “100%”, “inevitable”, and “imminent” (4829, 1385, 5365). However, among the community members who had left after the floods, almost all explicitly described reduced feelings of fear and anxiety during rainfall after relocating. However, while moving alleviated some of participants’ fears, it was not an easy or unequivocal decision. For example, a participant describes their mixed feelings about relocating:

“I have a lot of mixed feelings about it. I was really sad to lose my town. I was really sad to have to walk away. I really felt defeated… On the other hand, I was very grateful and I realize that I’m really lucky… not having that ongoing worry. We can sleep through the night when it rains. Whereas we never could before [we moved away].”(8503)

#### 3.2.3. Theme 3: Financial Burdens

Small local business owners in the Historic District described losing “$60,000 worth of equipment and materials.” (6385) Many business owners had “spent their life savings or [their] retirement funds and whatever little money [they] had” to reopen their businesses, only to be “hit with an even more devastating flood” (8303) less than two years later. The second flood left many small business owners unable to reopen due to the financial losses. Some business owners “lost their lease [and were] kicked out” (2847) due to changes in building ownership because of financial losses. One participant said, “people only have so much money to invest in a business, and when it washes away and the insurance doesn’t pick it up, they move on.” (7139). A couple business owners relocated their businesses outside of the flood zone, but most did not reopen at all.

## 4. Discussion

Though themes presented in the results section may have manifested in different ways among participants, participants did not outwardly express beliefs or opinions contrary to these themes. However, as we can see from our results, some of the reasons participants gave for staying were also given by others for leaving. For example, for many participants, the community members were a reason for why they stayed because of the empathy, understanding, and flood risk security they provided. However, for other participants, seeing their beloved community members experience repetitive hardship each time they were flooded led to emotional exhaustion and a desire to distance themselves, and thus was a reason for why multiple participants left Ellicott City’s flood prone areas. As another example, financial burdens were a reason why some participants stayed and a reason why others left.

As we can see from the results of this study, the reasons why people stay, or leave is a complex nexus that includes community/historical, environmental, emotional, and economic components. These components were similarly seen in a study in Austria about the factors that contribute to why a household stayed in or relocated from a flood risk area [34]. Consistent with these previous findings [34], our results showed that that revealed the complexity of factors that contribute to relocation/non-relocation and support claims for the need for tailored responses based on unique personal combinations of community/historical values, environmental implications, emotional needs, and financial necessities. In the following subsections we describe some of the community/historical, environmental, emotional, and economic factors that may have impacted whether participants stayed or relocated.

In 2019, the Howard County government announced it’s Safe and Sound plan, a flood control project, for Ellicott City to financially support business and property owners, prepare for climate change, and involve the community [26]. At the end of 2020, Ellicott City’s Department of Planning and Zoning announced its Watershed Master Plan [35]. This plan intends to build on the Safe and Sound plan to provide more flood mitigation. Like the Safe and Sound plan, it also focuses on community building and attempting to preserve most historical buildings [26,35]. In this discussion section, we also comment on how the Master Plan may impact those who had been affected by the flash floods.

### 4.1. Community/Historical

A previously published study in India revealed that many individuals who experienced flooding preferred to create a community or family based emergency plan in case of a future flood rather than relocate [14]. Individuals who experienced flash flooding in Ellicott City described their community as a support network, a source of flood safety and security, and a reason for why they stayed. Similar to the participants in previous literature [14], the participants who experienced flash flooding in Ellicott City that stayed expressed having faith in their community’s support if another flood were to occur. Our results support the importance of the community for these individuals who experienced flash flooding, especially for those who stay.

Throughout our study interviews, we noticed how important the preservation of historical buildings and sense of community were for many participants, especially for participants who stayed in the flooded areas. This is an attribute particularly relevant to our study site. We suspect that the Safe and Sound plan and Master plan’s focus on community building and historical preservation may help decrease strain within the flooded community, especially on those who stayed in Ellicott City. Sense of community and connection to place can be important factors in how people respond to flooding disasters. This can relate not only to interpersonal connections but also links to historical buildings.

### 4.2. Environmental

In our study, over 15% of participants reported feelings of frustration regarding the potential exacerbation of flash flooding due to climate change. Over 40% of participants reported similar feelings of frustrations towards the local government and developers when participants’ efforts to stop impervious development that may cause further flash flooding were ignored. The Master plan in Ellicott City includes descriptions of the intention to continue development to make Ellicott City an economic hub for tourism [35]. This development may include intentions to address flood mitigation, but may also add impervious surfaces [35]. Based on the results of our study, multiple participants relocated away from Ellicott City due to the frustration related to continued development. Amongst those who stayed, many continued to express this same frustration and continued fear of future flash flooding. This aligns with existing literature revealing that fear of uncertainty regarding future flooding along with the expectation that floods may become more severe due to climate change and urbanization was a largely contributing reason for relocation [34,36].

### 4.3. Emotional

Almost all participants felt confident that future flash flooding would occur in Ellicott City and many of those who stayed expressed continued fear, especially during rainfall. However, participants who left reported feelings of relief after relocating. Previous studies revealed that flooding is associated with post-traumatic stress disorder and anxiety [37,38,39,40,41,42,43]. Though our particular study did not only focus on the mental health impacts of flash flooding, the findings from previous studies on mental health and flooding align with participants’ reports of rain induced fear and anxiety. However, neither Safe and Sound nor the Master plan include professional assistance for participants who have been psychological and emotionally impacted by the flash floods [26,35]. Participants, especially those who stayed, reported receiving emotional support from fellow individuals of the community who experienced flash flooding. As such, we believe that future research is needed to understand the benefits of support from relatable community members in comparison to professional support after flash flooding.

### 4.4. Economic

Previous literature reveals financial burdens associated with staying in a flood prone area due to factors such as repetitive repairs, healthcare costs, or work-related difficulties [44]. Other literature indicates the financial burdens of relocating including costs of a new home or uncertain employment opportunities [34,45]. Our study supports findings from both literature that associates financial burdens with staying and those that associate financial burdens with relocating [34,44,45]. Though almost all our participants in our study had flood insurance, individuals who experienced flash flooding may be constrained to stay unwillingly and others to leave unwillingly, by several factors related to insurance, employment, and/or costs of rebuilding, as evidenced through quotes by the participants.

### 4.5. Limitations and Advantages

Limitations of our study include selection bias. The individuals who were most impacted by the flooding in Ellicott City, know many other participants (snowball sampling), or were especially interested in flash flooding may have been more likely to participant in the study. Additionally, those who moved from Ellicott City or were no longer involved with the community may have been less likely to volunteer to participate. Our study may generalize the experience of individuals who were impacted by multiple floods. However, each individual as well as members of different communities may have different experiences based of varying characteristics such as socioeconomics or cultural values. For this study, we did not gather detailed information on the participants’ socioeconomic characteristics, but population characteristics (e.g., race, age, income) may be relevant in how people experience flooding disasters and warrant further research. The study included detailed interviews with 19 participants, however this sample size is within the recommended range for phenomenological research [28,29,30].

Advantages to our research study include aiding the understanding of qualitative factors that are difficult to measure, such as sense of place and sense of community. There exist very few studies about flash flooding, especially in the United States, and our unique study site that greatly values historical preservation and sense of community also adds to the novelty of the study.

## 5. Conclusions

In conclusion, our study revealed that the reasons for why individuals who experienced flash flooding stay, or leave may have community/historical, environmental, emotional, and economical components. This reveals the complexity of sense of place and relocation after natural/environmental disasters. These findings support claims from previous literature that suggest that sense of place is largely individualized based on unique history, memories, values, emotions, beliefs, and experiences associated with the place [13] and for the need to tailor response efforts based on an individual’s unique set of burdens [34]. With this paper, we do not intend to encourage individuals who experience flash flooding to either stay or leave the flood zone, but instead, we hope that this paper will help identify and aid understanding of burdens to help policy makers improve flood response efforts.

## Figures and Tables

**Figure 1 ijerph-19-10636-f001:**
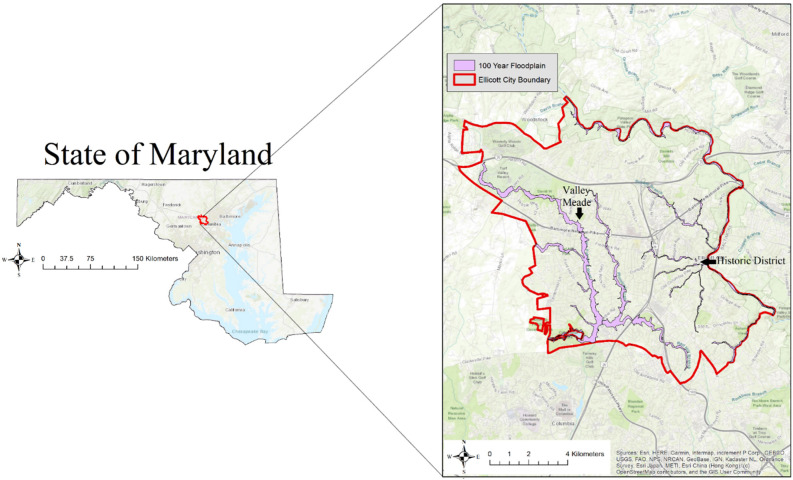
Our study site Ellicott City, MD and the 100-year floodplain (area with at least a 1% chance of flooding in a given year). Most participants either live/d or work/ed in the Historic District or Valley Meade during the flash flood(s). Maryland state boundary acquired from the US Census Bureau. Map was modified from Ellicott City Floodplains and Flood Zones [27].

**Table 1 ijerph-19-10636-t001:** Summary of participants.

Description	Number of Participants
Total	19
Experienced Both Floods	16
Relocated/no longer works in Ellicott City after 2016 flood	4
Relocated/no longer works in Ellicott City after 2018 flood	2
Worked AND lived in Ellicott City during flood(s)	5
Lived in Ellicott City flood zone during flood(s)	16
Still lives in Ellicott City flood zone	14
No longer lives in Ellicott City flood zone	2
Worked in Ellicott City flood zone during floods(s)	8
Still works in Ellicott City flood zone	4
No longer works in Ellicott City flood zone	4

## Data Availability

Data from this study are not publicly available.

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
