# Peer review of "Why They Stayed and Why They Left—A Case Study from Ellicott City, MD after Flash Flooding"

_ijerph, 2022, doi:10.3390/ijerph191710636_

Round 1

Reviewer 1 Report

This paper contributes interesting new qualitative data on the important issue of migration (or not) decisions following multiple flash flood hazards in the United States.

- Sense of Place is in the abstract and key words, and features in the stated contributions of the work. However, sense of place is not defined or discussed in the introduction/literature, and the introduction does not use any of the literature on sense of place & flood hazard specifically or natural hazards generally. In fact, theory on sense of place can help integrate all of the non-economic themes of the results & discussion.

- I would love to see a reference map accompanying the "study site" section, illustrating key features & processes described in this section

- Regarding sampling method & outcomes: I would like to see a detailed breakdown the frequencies of participants in terms of having business, residence, or both in flood zone, presence for one or both floods, and outcomes (rebuilding home, relocating home, rebuilding business, relocating business, closing business). I think it's misleading to calculate or discuss percentages at all for such a small unrepresentative sample. Did the authors make any special efforts to contact / engage folks who had moved? Did they find them though the social media networks directly or through other participants?

- define the "flood zone". Is this a known extent of flooding from one or both events? is it the FEMA 100-year or 500-year floodplain?

- can the authors include any statistics for these disasters to help contextualize the small sampling of interviews? e.g. # or % of people moving out of the flood area after each disaster?

- If the safe and sound plan and master plan are an important component of this research, they should be introduced in the introduction.

Author Response

Reviewer’s comment: This paper contributes interesting new qualitative data on the important issue of migration (or not) decisions following multiple flash flood hazards in the United States.

Response: Thank you!

Reviewer’s comment: Sense of Place is in the abstract and key words, and features in the stated contributions of the work. However, sense of place is not defined or discussed in the introduction/literature, and the introduction does not use any of the literature on sense of place & flood hazard specifically or natural hazards generally. In fact, theory on sense of place can help integrate all of the non-economic themes of the results & discussion.

Response: Thank you for the suggestion. We added text to the introduction (see lines 40-44) to describe sense of place. We also added a sentence in the conclusion section (lines 382-384) to connect some previous literature on sense of place to our findings.

Reviewer’s comment: I would love to see a reference map accompanying the "study site" section, illustrating key features & processes described in this section.

Response: Thank you for the suggestion. We added Figure 1, a map of our study site, to the paper. This map shows the 100-year floodplain.

Reviewer’s comment: Regarding sampling method & outcomes: I would like to see a detailed breakdown the frequencies of participants in terms of having business, residence, or both in flood zone, presence for one or both floods, and outcomes (rebuilding home, relocating home, rebuilding business, relocating business, closing business). I think it's misleading to calculate or discuss percentages at all for such a small unrepresentative sample.

Response: We’ve added more details to Table 1 per your suggestion. We also removed the percentages.

Reviewer’s comment: Did the authors make any special efforts to contact / engage folks who had moved? Did they find them though the social media networks directly or through other participants?

Response: We had 2 participants that relocated and 4 that stopped working in Ellicott city after the floods (See Table 1). A third participant stated that they were going to be relocating in the very near future (See Lines 151-153). We advertised the study on social media and used snowball sampling to gather participants. As such, we recognize that individuals who are no longer involved with the Ellicott City community would have been less likely to participate in this study. We acknowledged this in the limitations section (see lines 359-360).

Reviewer’s comment: define the "flood zone". Is this a known extent of flooding from one or both events? is it the FEMA 100-year or 500-year floodplain?

Response: Thank you for your suggestion. We created a map of Ellicott City and the 100-year floodplain (Figure 1) to help clarify.

Reviewer’s comment: can the authors include any statistics for these disasters to help contextualize the small sampling of interviews? e.g. # or % of people moving out of the flood area after each disaster?

Response: Thank you for the suggestion. We added that to Table 1. Table 1 shows the experience of the study participants and has % removed (as per the earlier suggestion). We also added text (lines 32-33) on the approximate number of people displaced after each of the floods.

Reviewer’s comment: If the safe and sound plan and master plan are an important component of this research, they should be introduced in the introduction.

Response: We only briefly mention the study site in the beginning of the introduction. We introduce the study site in the Methods section. We agree that this plan is important, so per your suggestion, we added this to the Study Site section as well (see lines 90-93).

Reviewer 2 Report

 1.      Line 67. Is it possible to show a study area map which is helpful for foreign readers to understand the site location? Where is the Ellicott City in Howard County, MD?

2.      Line 85. Please briefly describe participants’ attributes, ex: gender, education, age, ….

3.      Line 135. Table 1 shows that 87.5% of participants who lived in Ellicott City’s flood zone. How to define the flood zone? What is the flood return period for defining the flood zone?

4.      Line 151. What is the difference between bonds of community and sense of place? How about your definition for both terms?

5.      Line 186. Theme 3: Financial Burdens. Participants’ statements are too weak. Could you improve this paragraph?

6.      Line 231. Is there the flood insurance for participants? How to reduce the flooding risk?

7.      Line 275. What is the effect of participants’ attribute on social/cultural, environmental, emotional, and economic factors? This is an important point. Please explain it in detail.

8.      Line 308. “Almost all participants felt confident that future flash flooding would occur in Ellicott City and many of those who stayed expressed continued fear, especially during rainfall.” Is there flood control project in Ellicott City? Residents are continued fear about the flood. What is the role of local government for flood control?

Author Response

Reviewer’s comment: Line 67. Is it possible to show a study area map which is helpful for foreign readers to understand the site location? Where is the Ellicott City in Howard County, MD?

Response: Thank you for the suggestion. We added a map of Ellicott City (Figure 1) for clarification. This shows the 100-year flood plain.

Reviewer’s comment: Line 85. Please briefly describe participants’ attributes, ex: gender, education, age.

Response: All participants stated that they were at least 18 years of age, but we did not collect other socioeconomic information from participants. We present overall socioeconomic information for the study area (lines 74-80). To address the reviewer’s point we added text to the limitations section (lines 365-368).

Reviewer’s comment: Line 135. Table 1 shows that 87.5% of participants who lived in Ellicott City’s flood zone. How to define the flood zone? What is the flood return period for defining the flood zone?

Response: We added Figure 1 which shows the 100-year floodplain for clarification.

Reviewer’s comment: Line 151. What is the difference between bonds of community and sense of place? How about your definition for both terms?

Response: Thank you for your questions. We added more information on sense of place in the introduction section (lines 40-44). We hope this helps clarify the difference between sense of place and bonds of community.

Reviewer’s comment: Line 186. Theme 3: Financial Burdens. Participants’ statements are too weak. Could you improve this paragraph?

Response: We apologize. This Financial Burdens Theme 3 section in the results is simply quotes from participants. Since only a few participants mentioned financial burdens (lines 206-210 and 251-260), we did not have many quotes to include in this section. This is the reason this is Theme 3 rather than Theme 1, which is the theme that was mentioned most often by participants.

Reviewer’s comment: Line 231. Is there the flood insurance for participants? How to reduce the flooding risk?

Response: Yes, there is flood insurance. It was mentioned several times in the paper, but we added further mention of this to lines 347-348 in the discussion to clarify.

Reviewer’s comment: Line 275. What is the effect of participants’ attribute on social/cultural, environmental, emotional, and economic factors? This is an important point. Please explain it in detail.

Response: We apologize for any confusion. To avoid confusion in the future, we have renamed the social/cultural subsection to community/historical. We also added text (see lines 308-310) for clarification in this subsection.

Reviewer’s comment: Line 308. “Almost all participants felt confident that future flash flooding would occur in Ellicott City and many of those who stayed expressed continued fear, especially during rainfall.” Is there flood control project in Ellicott City? Residents are continued fear about the flood. What is the role of local government for flood control?

Response: Yes, there are flood control projects from the local government including those part of the Safe in Sound Plan (see lines 282-287 in discussion). We added text to clarify that this is a flood control project (line 286) that was led by the Howard County government (line 285).